# Phenotypic and Molecular Characterization of Commensal, Community-Acquired and Nosocomial *Klebsiella* spp.

**DOI:** 10.3390/microorganisms9112344

**Published:** 2021-11-12

**Authors:** Marta Gómez, Arancha Valverde, Rosa del Campo, Juan Miguel Rodríguez, Antonio Maldonado-Barragán

**Affiliations:** 1Department of Nutrition and Food Science, Complutense University of Madrid, 28040 Madrid, Spain; marta_gmz@hotmail.com (M.G.); jmrodrig@ucm.es (J.M.R.); 2Department of Microbiology, Hospital Universitario Ramón y Cajal IRYCIS, 28034 Madrid, Spain; aranchavalverde@gmail.com (A.V.); rosa.campo@salud.madrid.org (R.d.C.); 3Infection and Global Health Research Division, School of Medicine, University of St. Andrews, North Haugh, St Andrews KY16 9TF, UK

**Keywords:** *Klebsiella*, *rpoB*, virulence, siderophores, biofilms, antibiotic resistance, bacteriocins

## Abstract

*Klebsiella* spp. is a relevant pathogen that can present acquired resistance to almost all available antibiotics, thus representing a serious threat for public health. While most studies have been focused on isolates causing community-acquired and nosocomial infections, little is known about the commensal isolates colonizing healthy subjects. We describe the molecular identification and the phenotypic characterization of commensal *Klebsiella* spp. from breast milk of healthy women and faeces from healthy breast-fed infants, which were compared with isolates from community-acquired infections and from a nosocomial NICU outbreak. The phylogenetic analysis of a 454-bp sequence of the *rpoB* gene was useful for species identification (*K. pneumoniae*, *K. variicola*, *K. quasipneumoniae*, *K. oxytoca*, *K. grimontii*, *K. michiganensis*, *Raoultella planticola* and *R. ornithinolytica*), previously misidentified as *K. pneumoniae* or *K. oxytoca* by biochemical methods. Globally, we report that commensal strains present virulence traits (virulence genes, siderophores and biofilms) comparable to community-acquired and NICU-infective isolates, thus suggesting that the human microbiota could constitute a reservoir for infection. Isolates causing NICU outbreak were multi-drug resistant (MDR) and ESBLs producers, although an imipenem-resistant commensal MDR *K. quasipneumoniae* isolate was also found. A commensal *K. pneumoniae* strain showed a potent bacteriocin-like inhibitory activity against MDR *Klebsiella* isolates, thus highlighting the potential role of commensal *Klebsiella* spp. in health and disease.

## 1. Introduction

*Klebsiella* spp. are ubiquitous in nature and can be found in environment samples (surface water, sewage, soil and plants) and, also, colonizing the mucosal surfaces of healthy mammals [1]. The major species of this genus is *Klebsiella pneumoniae*, followed by *Klebsiella oxytoca*, both considered as opportunistic pathogens with major relevance in community- and hospital-acquired (nosocomial) infections, which are particularly severe in immunocompromised subjects such as those hospitalized in transplant, intensive care (ICU), or neonatal units (NICU) [2].

Phylogenetic analyses have shown that *K. pneumoniae* complex comprises seven phylogenetic groups (Kp1 to Kp7). Kp1 is the most abundant and includes *K. pneumoniae sensu stricto*; Kp2, Kp3, Kp4, Kp5, Kp6 and Kp7 include *K. quasipneumoniae* subsp. *quasipneumoniae*, *K. variicola* subsp. *variicola*, *K. quasipneumoniae* subsp. *similipneumoniae*, *K. variicola* subsp. *tropicalensis*, *K. quasivariicola* and *K. africanensis*, respectively [3,4,5]. On the other hand, the *K. oxytoca* complex comprises six phylogroups, Ko1, Ko2, Ko3, Ko4, Ko6 and Ko8, including *K. michiganensis*, *K. oxytoca sensu stricto*, *K. spallanzanii*, *K. pasteurii*, *K. grimonti* and *K. huaxiensis* [4,6].

Isolates grouped in the *K. pneumonie* complex possess similar biochemical and phenotypic features, being inaccurately identified as *K. pneumoniae sensu stricto* or as *K. variicola* by conventional microbiological methods. Thus, a recent study has shown that *K. variicola* and *K. quasipneumoniae*, which are often misidentified as *K. pneumoniae*, cause severe life-threatening infections similar to *K. pneumoniae* [7]. In the same manner, the members of the *K. oxytoca* complex are frequently misidentified as *K. oxytoca sensu stricto*. However, it has been suggested that *K. michiganensis*, which is commonly erroneously identified as *K. oxytoca*, is likely to be more clinically relevant than *K. oxytoca* in human-associated infections [8].

Commensal colonizing isolates have been far less studied than clinically relevant community-acquired and nosocomial *Klebsiella* spp. causing infectious diseases. In fact, studies about commensal *Klebsiella* isolates from healthy non-hospitalized subjects are currently very scarce. By using culture-independent methods, it has been estimated that approximately 3.8% of stool samples from healthy individuals contains *K. pneumoniae* [9]. In a recent study with healthy adults, it has been shown that although *Klebsiella* spp. constitute minor bacterial components of the human gut microbiota, some *K. pneumoniae* isolates could present a great potential to cause infections [10]. This was previously observed in a study with healthy Korean adults, where a high proportion of subjects showed faecal carriage of *K. pneumoniae* sequence type 23, which is associated with pyogenic liver abscess in Korea [11]. In fact, two recent studies with hospitalized adults have shown that gastrointestinal colonization with *K. pneumoniae* is strongly linked to subsequent infections in these subjects during hospitalization and demonstrated that a large proportion of *K. pneumoniae* infections were acquired from patient’s own microbiota [12,13].

Nosocomial *Klebsiella* infections are especially problematic in preterm neonates causing neonatal sepsis, including both early- and late-onset infections [14]. In a recent work, it has been shown that “healthy” antibiotic-treated preterm infants hospitalized in NICUs can harbour different *Klebsiella* spp. such as *K. pneumoniae*, *K. quasipneumoniae*, *K. grimontii* and *K. michiganensis*, which could greatly contribute to the resistome [8].

Overall, all these studies suggest that *Klebsiella* spp. is a habitual commensal in the healthy human microbiota, which could provide a potential reservoir for infection. In the light of these findings, the phenotypic and molecular characterization of *Klebsiella* isolates from healthy subjects could contribute to understanding the relevance of commensal *Klebsiella* spp. as a reservoir of potentially dangerous traits for human health. Previous works have determined that this genus can be a part of the human milk microbiota of healthy women [15], frequently arising from the use of pumps for milk expression [16]. Breastfeeding women are a representative sample of the general population since the majority of them will give birth at some point in a hospital, where the newborn will be ideally breastfeed within the first hour of birth. As a consequence, human milk represents one of the first vehicles for the mother-to-infant transfer of microbes [17].

In these studies, we obtained a collection of *Klebsiella* isolates from the milk of healthy women and from the meconium and faeces of breast-fed term infants. The aim of this study was to identify and characterize these commensal *Klebsiella* isolates and compare them with nosocomial isolates from an NICU outbreak and from community-acquired infections isolates.

## 2. Materials and Methods

### 2.1. Bacterial Strains and Growth Conditions

A total of 56 *Klebsiella* spp. isolates, which were initially identified as *K. pneumoniae* (*n* = 35) or *K. oxytoca* (*n* = 21) by routine biochemical methods (Wider system; Francisco Soria Melguizo, S.A., Madrid, Spain), were included in this study (Table 1). These isolates were obtained from different origins: (i) 20 commensal isolates from human milk (*n* = 5), meconium (*n* = 1) and faeces of breast-fed infants (*n* = 14) of healthy individuals from the bacterial collection of the research group 920080 (Complutense University of Madrid, Spain); (ii) 26 community-acquired isolates causing bacteraemia in adult outpatients at the Hospital Ramón y Cajal (Madrid, Spain); and (iii) 10 clinical isolates from the neonatal ICU of the Hospital 12 de Octubre (Madrid, Spain) from blood (*n* = 2), catheter (*n* = 1), environmental surfaces (*n* = 2) and colonizing the gut of newborns admitted at NICU (*n* = 5). The Ethical Committee on Clinical Research of the Hospital Clínico San Carlos of Madrid (Spain) approved the study protocol (reference 10/017-E). In the frame of such protocol, samples used to isolate the bacterial strains were obtained after informed written consent of each person or, when required, of the infants’ legal guardians. All strains were routinely grown in Brain Heart Infusion (BHI; Oxoid, Basingstoke, UK) broth, BHI solid medium (containing 1.5% *w*/*v* agar) and MacConkey agar medium (BioMèrieux, Marcy l’Etoile, Francia) at 37 °C for 24 h. *K. pneumoniae* subsp. *pneumoniae* DSMZ30104^T^, *K. pneumoniae* subsp. *rhinoscleromatis* DSMZ16231^T^, *K. pneumoniae* subsp. *ozaenae* DSMZ16358^T^, *K. pneumoniae* CECT 142, *K. pneumoniae* CECT 517 and *K. oxytoca* CECT 860^T^ were used as reference strains.

### 2.2. Molecular Identification of Isolates

All isolates were re-identified by sequencing a 454-bp fragment of the *rpoB* gene amplified by PCR with KrpoB-for and KrpoB-rev primer pair (Appendix A). PCR conditions were as follows: 1 cycle of 94 °C for 4 min, 30 cycles of 94 °C for 30 s, 55 °C for 30 s and 72 °C for 30 s and a final extension of 72 °C for 5 min. Amplicons were purified using the Nucleospin Extract II kit (Macherey-Nagel, Düren, Germany) and sequenced (ABI Prism 3730; Applied Biosystems, Foster City, CA, USA) at the Genomics Unit of the Universidad Complutense de Madrid (Madrid, Spain). The resulting sequences were used to search against reference sequences deposited in the EMBL database using BLASTn algorithm (http://www.ncbi.nlm.nih.gov; accessed on 7 July 2021). The identity of the isolates was determined on the basis of the highest scores. The *rpoB* gene sequences obtained were deposited in the GenBank database, under the accession numbers KJ499842 to KJ499903.

### 2.3. Phylogenetic Analysis

For phylogenetic analysis, the *rpoB* sequences obtained were aligned by using the Clustal W method [18] with the MEGA version 6 (Revision 6.06, update February 2014) software created by Tamura, Stecher, Peterson, Filipski, and Kumar [19]. Phylogenetic trees were constructed based on the neighbour-joining method with the Jukes–Cantor parameter model [20]. Bootstraping analysis (1000 replicates) was performed to study the stability of the groupings. The *rpoB* sequences of the following type species were obtained from the GenBank database and included in the phylogenetic analysis the following: *K. variicola* F2R9^T^ (AY367356), *Raoultella ornithinolytica* ATCC 31898^T^ (AF129447), *Raoultella planticola* ATCC 33531^T^ (AF129449) and *Staphylococcus sciuri* subsp. *carnaticus* ATCC 700058^T^ (DQ120748).

### 2.4. Genotyping of Isolates

Pulsed-field gel electrophoresis (PFGE) was performed as described previously [21]. Chromosomal DNA of each isolate was digested with 30 U of XbaI (TaKaRa Bio Inc, Shiga, Japan). Electrophoresis was carried out in a CHEF DR II apparatus (Bio-Rad, Laboratories, Hercules, CA, USA) with the following conditions: 14 °C, 6 V/cm^2^ and 10–40 s for 24 h. Dendrograms of genetic relationships were constructed by using the Phoretix 1D software (version 5.0; Nonlinear Dynamics Ltd., Newcastle upon Tyne, UK) based on the Dice coefficient.

Small plasmids were extracted from 16 h BHI broth cultures with the “QIAprep Spin Miniprep Kit” (QIAgen) as recommended by the manufacturer and visualized in a 1% agarose gel by using conventional electrophoresis (90 V for 90 min). Supercoiled ladder (1–16 kb) (Invitrogen, Paisley, UK) was used as molecular weight marker. Large plasmids (>16 kb) were extracted using PFGE-S1 nuclease digestion (Takara Bio Inc, Shiga, Japan) with the following conditions: 14 °C, 6 V/cm^2^ and 5–25 s for 3 h followed by 30–45 s for 12 h. Size was determined using the Lambda Ladder PFG Marker (48.5–1000, 18 Kb) and Low Range PFG Marker (0.13–194 Kb) (New England Biolabs, Inc.) as references. The resulting plasmid profiles were graphically represented and analysed using the software available on http://insilico.ehu.es/dice_upgma/ (accessed on 7 July 2021) to generate dendrograms by UPGMA clustering using Dice correlation.

### 2.5. Antimicrobial Susceptibility Testing

Minimal inhibitory concentrations (MICs) to antibiotics were evaluated by a microdilution method using the Sensititre EMIZA 9EF (TREK Diagnostic Systems, Cleveland, EEUU) plates following the manufacturer’s instructions. Production of Extended-spectrum β-lactamases (ESBLs) was tested by the double-disk synergy test [22] containing ceftazidime/ceftazidime plus clavulanate or cefotaxime/cefotaxime plus clavulanate. The presence of CTX-M β-lactamase-encoding genes (*bla*_CTX-M_) was identified by multiplex PCR using the oligonucleotides CTX-M-1G-F, CTX-M-1G-R, CTX-M-2G-F, CTX-M-2G-R, CTX-M-9G-F and CTX-M-9G-R (Appendix A) and conditions described previously [23]. P1 and P2b primers (Appendix A) were used to amplify *bla*_CTX–M_ subgroup I genes [24]. Multidrug-resistance (MDR) was defined as non-susceptibility (resistance) to at least one agent in three or more antimicrobial categories [25].

### 2.6. Virulence Determinants

Presence of the *magA*, *rmpA*, *wabG*, *uge*, *kfu* and *fimH* genes encoding potential virulence factors was determined by PCR using specific primers (Appendix A) and conditions described previously [26,27]. A novel multiplex PCR was designed to detect genes associated to biosynthesis or receptors of the siderophores aerobactin (*iutA*, *iucB*), enterobactin (*fepA*, *fepC*) and yersibactin (*fyuA*, *ybtT)*. For this purpose, six oligonucleotide pairs were designed (iutA-F/iutA-R, iucB-F/iucB-R, fepA-F/fepA-R, fepC-F/fepC-R, FyuA-F/FyuA-R and YbtT-F/YbtT-R; Appendix A), which result in the amplification of DNA fragments of 580, 692, 897, 280, 828 and 451 bp, respectively. PCR conditions were as follows: 1 cycle of 94 °C for 4 min, 30 cycles of 94 °C for 30 s, 62 °C for 30 s, 72 °C for 1 min and a final extension of 72 °C for 5 min.

### 2.7. Hypermucoviscosity, Biofilms, Siderophores and Bacteriocin Activity Assays

The hypermucoviscous phenotype was determined by the string test [28]. The biofilm formation ability was analysed in polyvinylchloride plastic (PVC) microtiter plates as described previously [29,30]. Siderophores production was quantified in cell-free supernatants [31]. The ability to inhibit the growth of other strains by production of bacteriocin-like substances was tested both on solid and broth medium, according to the direct [32] and the “spot-on-the-lawn” methods [33], respectively.

## 3. Results

### 3.1. Molecular Identification of Klebsiella Species

The amplification and sequencing of a 454 bp fragment of the *rpoB* gene achieved a great level of discrimination of the *Klebsiella* isolates at the species level when compared against the *rpoB* sequences from the validated type strains (Table 1). Among the 21 isolates initially identified as *K. oxytoca* by routine biochemical methods, just 1 isolate was identified as *K. oxytoca sensu stricto* by *rpoB* sequencing. The rest of the isolates were identified as *K. michiganensis* (*n* = 16), *K. grimontii* (*n* = 2), *K. pneumoniae* (*n* = 1) and *R. ornithinolytica* (*n* = 1). In the case of the 35 *Klebsiella* isolates initially identified as *K. pneumoniae* by biochemical methods, most of them were identified as *K. pneumoniae sensu stricto* (*n*= 29) based on *rpoB* sequencing; the remaining isolates were identified as *K. variicola* (*n* = 2), *K. quasipneumoniae* subsp. *similipneumoniae* (*n* = 1), *K. quasipneumoniae* subsp. *quasipneumoniae* (*n* = 1), *K. michiganensis* (*n* = 1) and *Raoultella planticola* (*n* = 1). The reference strains *K. pneumoniae* subsp. *pneumoniae* DSMZ30104^T^, *K. pneumoniae* subsp. *rhinoscleromatis* DSMZ16231T, *K. pneumoniae* subsp. *ozaenae* DSMZ16358^T^, *K. pneumoniae* CECT 142, *K. pneumoniae* CECT 517 and *K. oxytoca* CECT 860^T^ were correctly identified to the species level by *rpoB* sequencing.

### 3.2. Phylogenetic Analysis Based on rpoB

The phylogenetic analysis based on the partial amplification of the rpoB gene, supported the molecular identification of all Klebsiella isolates. Indeed, all *K. pneumoniae* isolates clustered with phylogenetic group KpI (*K. pneumoniae*), while isolates MV91-24 and MV91-42 clustered with KpII-A (*K. quasipneumoniae* subsp. *similipneumoniae*) and KpII-B (*K. quasipneumoniae* subsp. *quasipneumoniae*), respectively. The MV3-1 and K12-3 isolates clustered with KpIII (*K. variicola* subsp. *varicola*). All isolates identified as *K. michiganensis* clustered with phylogroup KoI (*K. michiganensis*), while isolate Ko7 clustered with KoII (*K. oxytoca*) and isolates Ko8 and Ko11 clustered with KoIV (*K. grimontii*). The two identified Raoultella isolates, Ko1 and Ko10, clustered with *R. planticola* and with *R. ornithinolytica*, respectively (Figure 1).

### 3.3. Genetic Diversity

A high genetic diversity among the *K. pneumoniae* and the *K. oxytoca* complex isolates was detected by PFGE (Appendix A). However, within the *K. pneumoniae* isolates, two clonal groups (>80% similarity) were identified: CP1, formed by *K. pneumoniae* K12-1, K12-6 and K12-8, and CP2, formed by *K. pneumoniae* K12-4 and K12-9 (Appendix A). All these clonal isolates were isolated from the NICU’s outbreak. In the *K. oxytoca* complex, we identified two clonal groups (CM1 and CM2), formed by *K. michiganensis* HA001 and HA009, and *K. michiganensis* HV1-02 and HV2-11, respectively (Appendix A). Those isolates were isolated from faeces from breast-fed healthy term infants.

Plasmid profiles showed that a total of 28 (89%) isolates from the *K. pneumoniae* complex (Appendix A) and 18 (85%) from the *K. oxytoca* complex (Appendix A) contained plasmids (1 to 8 plasmids, ranging from 1 to 600 kb).

### 3.4. Antimicrobial Susceptibility

Low antibiotic resistance rates were detected in both commensal and bacteraemic community-acquired collections (Figure 2). The isolates were susceptible to most antibiotics tested with the exception of *K. quasipneumoniae* subsp. *similipneumoniae* MV91-24, an MDR and imipenem-resistant isolate from faeces of a healthy breast-fed infant, and *K. pneumoniae* Kp9 and *K. michiganensis* Ko12, two MDR isolates from community-acquired infections.

All isolates from the NICU showed a MDR phenotype, including ESBL production, with the unique exception of *K. pneumoniae* K12-2 (Figure 2). The antibiotic susceptibility profiles of *K. pneumoniae* K12-1, K12-4, K12-6, K12-8, K12-9 and K12-10, *K. michiganensis* K12-5 and K12-7 and *K. variicola* K12-3 were compatible with ESBLs production, a fact that was subsequently confirmed by the double-disk synergy test and the presence of the bla_CTX-M-15_ gene.

### 3.5. Virulence Determinants, Hypermucoviscosity, Biofilms and Siderophores

The presence of the wabG, uge and kfu genes was detected in 38.2%, 38.2% and 20.6% of the isolates from the *K. pneumoniae* complex, respectively (Figure 3). The wabG gene was more represented among commensal than among community-acquired isolates (*p* < 0.05), while the uge gene was more abundant among the NICU outbreak isolates than among the community-acquired ones (*p* > 0.05). All members of the *K. pneumoniae* complex presented the fimH gene (with the exception of *K. pneumoniae* Ko9), whereas the magA and rmpA genes were not detected. In contrast, none of the isolates from the *K. oxytoca* complex contained any of the virulence genes studied by PCR.

A novel multiplex PCR targeting the biosynthesis and receptor genes of the siderophores aerobactin, enterobactin and yersibactin (Figure 3) showed that among the *K. pneumoniae* complex, 64.7% presented the fepA gene (enterobactin synthesis), 79.4% the fepC (enterobactin receptor), 14.7% the fyuA (yersibactin synthesis) and 14.7% the ybtT (yersibactin receptor). None of the isolates harboured the genes iutA and iucB encoding the synthesis and receptor genes of aerobactin. The fepC gene was more represented among commensal than among community-acquired isolates (*p* < 0.05).

Within the *K. oxytoca* complex, the fepA gene was not detected in any of the isolates, while the fepC gene was present in 55% of the isolates. The genes fyuA and ybtT were detected in 55% and 5% of the isolates. The fyuA gene was less represented among community-acquired than among commensal isolates (*p* < 0.05). The iucB gene encoding the aerobactin receptor was only detected in two isolates.

None of the Klebsiella isolates showed a hypermucoviscous phenotype. Only 14.7 % of the isolates from the *K. pneumoniae* complex (all belonging to *K. pneumoniae*) were able to grow on biofilms formation on PVC plates (Figure 3), while none of the isolates from the *K. oxytoca* group showed this property. No significant differences were found in the ability to produce siderophores when comparing the isolates from the *K. pneumoniae* and *K. oxytoca complex. However, commensal isolates produced more siderophores than community-acquired isolates (p > 0.05).*

### 3.6. Antimicrobial Activity

Antimicrobial activity was only produced by the commensal *K. pneumoniae* MV91-1 and the community-acquired *K. pneumoniae* Kp5 isolates (Figure 4). MV91-1 showed a wide inhibitory spectrum against the other Klebsiella isolates, being active against *K. pneumoniae*, *K. quasipneumoniae* subsp. *similipneumoniae*, *K. pneumoniae* subsp. *quasipneumoniae*, *K. variicola*, *R. ornithinolytica*, *K. grimontii* and *K. michiganensis* (Figure 3). In contrast, the inhibitory spectrum of Kp5 was narrower, being active only against *K. pneumoniae*, *K. variicola* and *K. michiganensis* (Figure 3). The inhibitory activity of both strains was displayed only on solid medium and was abolished after the addition of proteinase K (1 mg/mL final concentration).

## 4. Discussion

Species from the genus *Klebsiella,* such as *K. pneumoniae* or *K. oxytoca*, are well-known for their ability to cause a wide range of infections in humans, some of them with fatal consequences [1]. An additional concern is that *Klebsiella* spp. has readily developed antimicrobial resistance to multiple antibiotics, being difficult to treat and eliminate with current antibiotics, thus generating a serious threat to public health [34]. Although most studies have been focused on the study of *Klebsiella* isolates from clinical samples (nosocomial and community-acquired infections), recent studies have shown that our own gastrointestinal tract is a potential reservoir of *Klebsiella* isolates, making difficult the delimitation between pathogens and commensal isolates [10,11,12,13]. This, together with the increasing number of novel *Klebsiella* species associated with humans that could cause severe infections, requires the development of new studies focused on the identification and characterization of commensal *Klebsiella* isolates from healthy subjects. 

A proper identification of *Klebsiella* isolates at the species level is relevant from an epidemiological and clinical point of view, but routine biochemical tests have limitations since members of this genus possess similar biochemical and phenotypic features. Molecular identification based on ribosomal 16S rDNA gene sequences has been useful in defining bacterial relationships, including those of *Klebsiella* [35]; however, their value for delineating closely related species seems limited because of the scarce nucleotide variation. The identification of *Klebsiella* species and phylogenetic groups within *K. pneumoniae* and *K. oxytoca* based on molecular methods can now be reliably achieved based on the sequencing of housekeeping genes such as *rpoB*, *gyrA* and *parC* [4,5,36]. 

In this work, we have shown that the amplification and sequencing of a 454 bp fragment of the *rpoB* gene achieved a great level of discrimination for the identification of *Klebsiella* isolates, which was supported by the phylogenetic analysis, thus providing a fast and reliable tool to identify species belonging to this genus. Among the 35 isolates identified as *K. pneumoniae* by biochemical methods, 29 were identified as *K. pneumoniae sensu stricto* based on *rpoB*, while the remaining 6 corresponded to *K. variicola* (*n* = 2), *K. quasipneumoniae* subsp. *similipneumoniae* (*n* = 1), *K. quasipneumoniae* subsp. *quasipneumoniae* (*n* = 1), *K. michiganensis* (*n* = 1) and *R. planticola* (*n* = 1) (Table 1). Although most *K. pneumoniae* were correctly identified by routine biochemical approaches, the misidentification of *K. variicola* and *K. quasipneumoniae* as *K. pneumoniae* is according to the results provided by previous studies [5,7,8,37]. This should be carefully considered since *K. variicola* and *K. quasipneumoniae* can cause severe life-threatening infections similar to *K. pneumoniae* [7,8]. On the other hand, most of the isolates identified by biochemical methods as *K. oxytoca* (16 of 21) were identified as *K. michiganensis* based on *rpoB*, while the remaining 5 were classified as *K. grimontii* (*n* = 2), *K. pneumoniae* (*n* = 1), *K. oxytoca sensu stricto* (*n* = 1) and *R. ornithinolytica* (*n* = 1). Thus, just 1 of the 21 isolates identified as *K. oxytoca* by routine biochemical approaches was *K. oxytoca sensu stricto*, while the vast majority belonged to *K. michiganensis*. These results support the findings of a recent study that have suggested that *K. michiganensis* could be more clinically relevant than *K. oxytoca* in human-associated infections [8]. This misidentification could have hidden the actual clinical and epidemiological significance of these species and, therefore, the implementation of alternative taxonomic methods, such as the sequencing of the *rpoB* gene, should be encouraged in clinical microbiology laboratories. 

Interestingly, we found a great variety of commensal *Klebsiella* spp. isolated from healthy subjects (milk from lactating women and faeces from breast-fed infants), such as *K. pneumoniae*, *K. quapsineumoniae* subsp *similipneumoniae* and *K. quapsineumoniae* subsp *quasipneumoniae*, *K. variicola* and *K. michiganensis*. This is in line with a recent study where it has been described that “healthy” antibiotic-treated preterm infants hospitalized in NICUs can harbour different *Klebsiella* species, including *K. pneumoniae*, *K. quasipneumoniae*, *K. grimontii* and *K. michiganensis* [8]. Similarly, *K. pneumoniae* and *K. oxytoca* have been described as components of the human gut microbiota from healthy adults [9,10,11]. With respect to community-acquired infections, we also found a great variety of species, such as *K. pneumoniae*, *K. oxytoca*, *K. michiganensis*, *K. grimontii*, *R. planticola* and *R. ornithinolytica*. Although Raoultella spp. are infrequent human pathogens, *R. ornithinolytica* is considered as emerging bacteria causing human infections [38,39]. The low prevalence of *R. ornithinolytica*- and *R. planticola*-related infections could be due to their misidentification as *K. pneumoniae* or *K. oxytoca* by conventional biochemical tests. In relation to the isolates from the NICU outbreak, we found two *K. michiganensis* strains (K12-5 and K12-7) that had been misidentified as *K. pneumoniae* and *K. oxytoca*, respectively. A recent study reported, for the first time, an NICU nosocomial outbreak caused by MDR and ESBL-producing *K. michiganensis* isolates [40]. This could indicate that hospital outbreaks caused by *K. michiganensis* could have passed undetected due their misclassification as other *Klebsiella* species. In addition, we found that an NICU’s isolate (K12-3) previously misidentified as *K. pneumoniae* actually belonged to *K. variicola,* which is currently considered as an emerging human pathogen [7,41]

A high genetic diversity was found among the commensal and the community-acquired *Klebsiella* isolates, whereas those from the NICU seemed to be the result of a polyclonal dissemination. Globally, the commensal and community-acquired *Klebsiella* isolates were sensitive to most of the antibiotics assayed in this study, with the exception of the commensal MV91-24 (*K. quasipneumoniae* subsp. *similipneumoniae*), and the community-acquired Ko12 (*K. michiganensis*) and Kp9 (*K. pneumoniae*), which were multi-drug-resistant (MDR). In contrast, 9 out of the 10 *Klebsiella* isolates from the NICU were MDR, thus supporting the situation of nosocomial outbreak. In addition, the NICU *bla*_CTX-M-15_-producing *K. pneumoniae* K12-9 and the commensal *K. quasipneumoniae* MV91-24 isolates exhibited resistance to imipenem, thus indicating the wide dispersion of carbapenemases, either combined with ESBL or not, among different environments, a fact that has been described previously [42,43]. 

ESBL-producing *K. pneumoniae* can cause infection outbreaks in NICUs. In this study, some of the strains were isolated from blood cultures (*K. pneumoniae* K12-1 and K12-2), vascular catheters (*K. pneumoniae* K12-6), faeces (*K. michiganensis* K12-5 and K12-7 and *K. pneumoniae* K12-8, K12-9 and K12-10) or the environment (*K. variicola* K12-3 and *K. pneumoniae* K12-4) during the investigation of a neonatal-ICU outbreak. Most of them were MDR and ESBL-producing strains, despite the fact that they belonged to different *Klebsiella* species, thus suggesting the existence of a polyclonal outbreak in the NICU. The six *K. pneumoniae* isolates *bla*_CTX-M-15_ producers, were grouped in three phylotypes (K12-1, K12-6 and K12-8; K12-4 and K12-9; K12-10), while the two MDR *K. michiganensis* isolates (K12-5 and K12-7) were not genetically related (Appendix A). Thus, although many outbreaks have been associated with the dissemination of a single clone, the polyclonal dissemination of different ESBL-producing *Klebsiella* strains can happen in the same unit at the same time [44]. Plasmids can play an essential role in the dissemination of antibiotic resistance among clinically relevant pathogens, as it has been described previously [44,45,46]. In this study, we have described that *Klebsiella* isolates possess a variable number (from 0 to 8) of large and small plasmids ranging from 1 to 600 kb in size which could reflects its ability to acquire and transfer genetic determinants, including virulence genes, pathogenicity islands and antimicrobial-resistance genes. The epidemiology of the NICU outbreak would require further investigation by using genomic approaches, as it was not the primary scope of this work.

The inverse correlation between antibiotic resistance and virulence has been previously proposed [47,48]. In this study, however, we did not find any correlation between antibiotic resistance and the carriage of virulence genes or the ability to produce biofilm or siderophores, despite a heterogeneity in the virulence factors being reported by other authors [37]. In the same manner, we did not observe strong correlations between the source of isolates (commensal, community acquired or NICU outbreak) and the presence of virulence traits. Similarly, little differences were found in the phenotypic and genomic characteristics of *Klebsiella* isolates recovered from healthy and sick infants [8]. 

None of the *Klebsiella* isolates harboured the *magA* and *rmpA* genes, which have been associated with the hypermucoviscosity phenotype in liver-invasive strains [26,27]. This was in agreement with the phenotypic assays performed in this work since none of the *Klebsiella* isolates studied showed a hypermucoviscous phenotype. In contrast, all members of the *K. pneumoniae* complex (with the exception of *K. pneumoniae* Ko9) presented the *fimH* gene encoding the type 1 fimbrial adhesion. Thus, 100% of commensal *K. pneumoniae* strains that were isolated from healthy subjects harboured the *fimH* gene, which is in contrast with the results of other authors, who found that *K. pneumoniae* strains isolated from the gut microbiota of healthy subjects were negative for that gene [10]. Other studies, however, have described a similar proportion of the *fimH* gene in *K. pneumoniae* isolates from community-acquired bacteraemia [27,37,49]. 

Within the *K. pneumoniae* complex, the *wabG* gene, encoding proteins involved in lipopolysaccharide synthesis [50] was detected in 38.2% of isolates and was more represented among commensal than among community-acquired isolates. Interestingly, the *wabG* gene was present in nearly all strains that had been isolated from faeces. The *uge* gene, encoding uridine diphosphate galacturonate 4-epimerase [51], was detected in 38.2% of isolates and was more abundant among the NICU outbreak isolates than among the community-acquired ones. In a recent study on *Klebsiella* spp. causing community-acquired infections, it has been described a positive rate of 28.2% and 61.5% for *wabG* and *uge* genes, respectively [37], while in other studies, the rates of positivity were close to 100% for both *uge* and *wabG* among virulent clones of *K. pneumoniae* [49,52]. Neither *wabG* nor *uge* were present in the isolates belonging to the *K. oxytoca* complex. 

Iron is essential for the growth of most bacterial pathogens, so the ability to acquire iron is frequently associated with bacterial virulence. The *kfu* (iron-uptake system) gene has been associated with higher virulence in *K. pneumoniae* [26,53,54]. In our study, a low proportion of *K. pneumoniae* isolates carried the *kfu* gene, similarly as was recently described by other authors [10,37]. Interestingly, the two *K. variicola* strains (MV3-1 and K12-3) and *R. ornithinolytica* Ko10 presented the *kfu* gene. In addition, bacteria can obtain iron from the host by removing it via siderophore-mediated uptake systems. In our study, we designed a novel multiplex PCR to detect genes associated to biosynthesis or as receptors of the siderophores aerobactin (*iutA*, *iucB*), enterobactin (*fepA*, *fepC*) and yersibactin (*fyuA*, *ybtT*). Within the *K. pneumoniae* complex, the gene encoding enterobactin (*fepA*) was less abundant (64%) than the gene *fepC* encoding the receptor of enterobactin (79%). In addition, the enterobactin receptor gene (*fepC*) was more represented in commensal than in community-acquired isolates. In line with our results, recent studies have shown that enterobactin genes are present in a high proportion of the *K. pneumoniae* strains isolated from healthy and diseased preterm infants [8] and from community-acquired cases [37]. In relation to yersiniabactin, the synthesis (*fyuA*) and receptor (*ybtT*) genes were less represented than enterobactin-related genes, but all isolates harbouring the *fyuA* gene also presented the *ybtT* gene. In contrast, none of the isolates harboured the genes *iutA* and *iucB* encoding the synthesis and receptor genes of aerobactin. Those results agree with other studies were the yersiniabactin and aerobactin genes were less represented than those corresponding to enterobactin [10,37]. None of the *K. quapsineumonaie* isolates harboured any of the siderophores genes tested, while only one isolate of *K. variicola* harboured the gene encoding the enterobactin receptor.

Within the *K. oxytoca* complex, the *fepA* gene encoding enterobactin was not detected in any of the isolates, while the *fepC* gene encoding the enterobactin receptor was present in 55% of the isolates, all belonging to *K. michiganensis* strains. The gene encoding yersinibactin (*fyuA*) was more represented in the *K. oxytoca* that in the *K. pneumoniae* complex and was only harboured by *K. michiganensis* and *K. grimontii*. However, all isolates harbouring the gene encoding yersiniabactin lacked the gene encoding its corresponding receptor (*ybtT*), which was only detected in *K. oxytoca sensu stricto*. Interestingly, the *fyuA* gene was less represented among community-acquired than among commensal isolates (*p* < 0.05). Aerobactin was not detected in any of the isolates of the *K. oxytoca* complex, while its receptor (*iucB*) was only detected in two *K. michiganensis* isolates. The genomic analysis of *K. michiganensis* and *K. grimontii* strains isolated from preterm infants has shown the presence of the enterobactin and yersiniabactin genes and the absence of the aerobactin gene in these species [8].

Our study has shown that most of *Klebsiella* isolates would be able to produce enterobactin, while a much smaller percentage would produce either aerobactin or yersiniabactin. Although these results agree with previous studies [1,8,10,37], our work revealed the high proportion of *Klebsiella* isolates that possess the gene encoding the receptor for enterobactin but not the gene for its biosynthesis. These isolates would be able to cheat the siderophore enterobactin produced by other bacteria, which could increase their competitiveness [55]. Finally, we did not find any correlation between the phenotypic production of siderophores and the presence of siderophore genes. This could be due to the presence of other iron-uptake systems different from that reported in this study, the lack of functionality of some of the genes studied by PCR or the existence of regulatory mechanisms, which would require further investigation. 

The ability of the *Klebsiella* to produce antimicrobial compounds both in solid and liquid medium was also addressed in this work. Two isolates of *K. pneumoniae*, MV91-1 (commensal) and Kp5 (community-acquired), inhibited the growth of other *Klebsiella* strains when assayed on solid medium (Figure 4). The inhibitory activity of both strains was sensitive to proteinase K, thus indicating that they most likely produce bacteriocins, which are traditionally defined as antimicrobial compounds of proteinaceous nature with antimicrobial activity against related species [56]. The inhibitory spectrum of *K. pneumoniae* MV91-1 against the collection of *Klebsiella* isolates included in this work was extraordinarily wide, being active against *K. pneumoniae*, *K. quasipneumoniae* subsp. *similipneumoniae* and subsp. *quasipneumoniae*, *K. variicola*, *R. ornithinolytica*, *K. grimontii* and *K. michiganensis*. In contrast, the inhibitory spectrum of *K. pneumoniae* Kp5 was narrower, being active only against *K. pneumoniae*, *K. variicola* and *K. michiganensis*. Interestingly, both *K. pneumoniae* MV91-1 and Kp5 were active against all the MDR and ESBL-producing isolates from the NICU’s outbreak. In addition, *K. pneumoniae* MV91-1 was active against the imipenem-resistant *K. pneumoniae* K12-9 and MV91-24 isolates. The characterization of these *K. pneumoniae* bacteriocins would open the door to the development of new antimicrobial strategies to combat *Klebsiella* strains, thus reducing the burden of antimicrobial resistance. In a recent study, it has been shown that bacteriocins from *Klebsiella* can be used for broad and efficient control of *Klebsiella* pathogens, in particular against MDR isolates [57].

The presence of different *Klebsiella* spp. in the milk of healthy women and in faeces of healthy breast-fed infants [15,16,17] suggest that these species are natural inhabitants of human microbiota, as has been described previously [8,9,10,11,12,13]. Attending to the virulence factors detected, the production of siderophores and biofilms and the plasmid content (small and large plasmids) observed, commensal strains were virtually indistinguishable from community-acquired and NICU outbreak isolates. Thus, these commensal isolates have all the potential to become pathogenic and cause infectious diseases, which is line with the recent studies showing that many infections are self-acquired from the patient’s own gastrointestinal microbiota [12,13]. The presence of virulence traits within each commensal strain could be determined by some of the conserved genes that form the core genome within each Klebsiella species, while other virulence and antibiotic resistance genes would be determined by a pool of accessory genes (accessory genome) that can be shared between different Klebsiella species and even between different genera [58]. Nowadays, it is becoming more evident that Klebsiella has the ability of exchanging and assembling a wide portfolio of genes that are involved in colonization, infection and antimicrobial resistance, which in the last instance would determine if a commensal strain remains asymptomatic or turns pathogenic [58]. Commensal Klebsiella isolates, which are well adapted to live and compete within the human microbiota, are prone to acquiring antibiotic resistant genes, by horizontal gene transfer, after the colonization with antibiotic-resistant Klebsiella strains acquired in hospital environments, such as an NICU, or from other host bacteria that have developed resistance after antibiotic treatments [59]. This combination (adaptation to the host and antibiotic resistance) could lead to the apparition of new MDR *Klebsiella* isolates that could be difficult to treat with current antibiotic-based strategies. This could be the case of the commensal isolate *K. quapsineumoniae* subsp. *similipneumoniae* MV91-42 (isolated from a healthy breast-fed infant) which is MDR, ESBL-producing and resistant to imipenem and that deserves further investigation. 

Our results highlight the importance of studying *Klebsiella* spp. from the microbiota of healthy people, which could help to create surveillance programs aimed to know the prevalence of antibiotic resistance in the population. In this line, it has been proposed that the screening of the patient’s gut microbiota after their admission to hospital could help to guide the application of the correct treatment [60]. Finally, we should increase our efforts in the development of novel alternatives or complementary strategies to antibiotics, which could help to reduce AMR in bacteria. These strategies could include the use of live biotherapeutic products, which has been demonstrated to be a real alternative to the use of antibiotics to combat infectious diseases [61] or the use of bacteriocins, such as plantaricin NC8 [62], as adjuvant in combination therapy to potentiate the effects of antibiotics and reduce their overall use [63]. The fact that the commensal isolate *K. pneumoniae* MV91-1 was able to inhibit the growth of the MDR, ESBL-producing, imipenem-resistant strains K12-9 and MV91-24, deserves to be further explored.

## 5. Conclusions

By sequencing a short region of the *rpoB* gene, we have determined that *K. variicola* and *K. pseudoneumoniae* isolates could be misidentified as *K. pneumoniae* by routine biochemical methods and that this biochemical misidentification is especially relevant within the *K. oxytoca* complex, where most isolates identified as *K. oxytoca* belong to *K. michiganensis* and, to a lesser extent, to *K. grimontii* and *Raoultella* species. Attending to the presence of virulence determinants, the production of siderophores or the ability to form biofilms, we were unable to distinguish between commensal *Klebsiella* strains isolated from healthy subjects and *Klebsiella* strains isolated from community-acquired infections or NICU nosocomial outbreaks. The only difference observed was the antibiotic susceptibility profiles since the strains isolated from the NICU’s outbreak were ESBLs producers and harboured CTX-M-15 genes. Finally, while this study reveals that the human microbiota could constitute a reservoir of commensal *Klebsiella* isolates with the potential to become pathogenic, we have also demonstrated that it could be envisaged as a potential source of novel antimicrobials to combat the increasing threat of antibiotic resistance in *Klebsiella* spp.

## Figures and Tables

**Figure 1 microorganisms-09-02344-f001:**
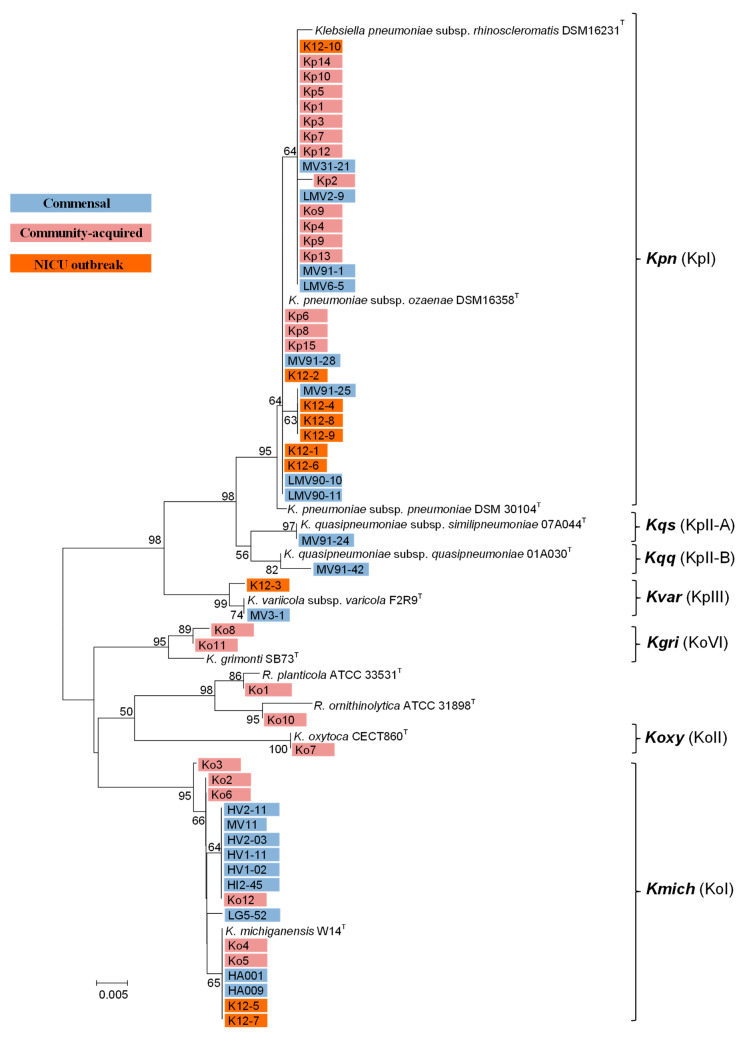
Phylogenetic relationships of 56 *Klebsiella* isolates and reference type strains by comparison of partial *rpoB* gene sequences (454 bp): The tree was based on the neighbour-joining method, using the Jukes–Cantor parameter model. Numbers on the tree indicate bootstrap values calculated for 1000 subsets for branch-points greater than 50%. Bar, 0.005 nucleotide changes per nucleotide position. Phylogenetic groups (KpI, KpII-A, KpII-B, KpIII, KoII, KoVI and KoI) are shown in brackets (Kpn: *K. pneumoniae*; Kqs: *K. quasipneumoniae* subsp. *similipneumoniae*; Kqq: *K. quasipneumoniae* subsp. *quasipneumoniae*; Kvar: *K. variicola*; Kgri: *Klebsiella grimontii*; Koxy: *K. oxytoca*; Kmich: *K. michiganensis*).

**Figure 2 microorganisms-09-02344-f002:**
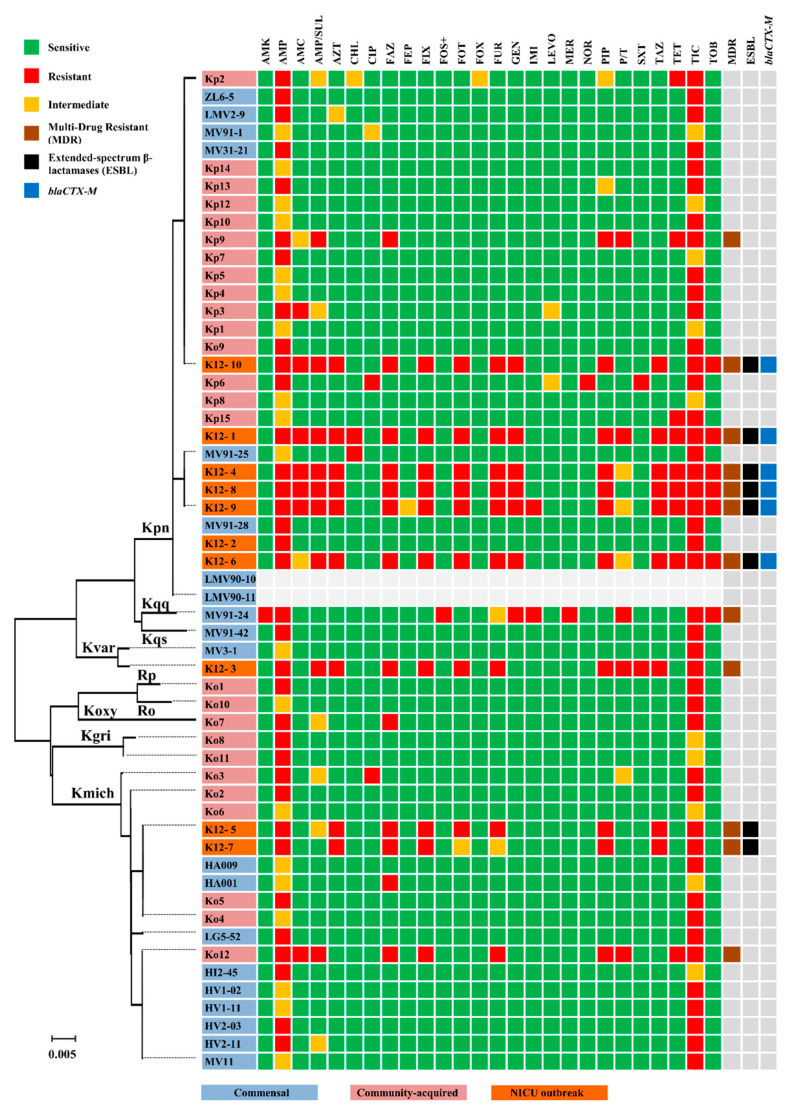
Antibiotic resistance profiles among the Klebsiella isolates analysed in this study: Antibiotics: amikacin (AMK), ampicillin (AMP), amoxicillin–clavulanic acid (AMC), ampicillin-sulbactam (AMP/SUL), aztreonam (AZT), chloramphenicol (CHL), ciprofloxacin (CIP), cefazolin (FAZ), cefepime (FEP), cefixime (FIX), fosfomycin (FOS), cefotaxime (FOT), cefoxitin (FOX), cefuroxime (FUR), gentamicin (GEN), imipenem (IMI), levofloxacin (LEVO), meropenem (MER), norfloxacin (NOR), piperacillin (PIP), piperacillin-tazobactam (P/T), trimethoprim-sulfamethoxazole (SXT), ceftazidime (TAZ), tetracycline (TET), ticarcillin (TIC) and tobramycin (TOB).

**Figure 3 microorganisms-09-02344-f003:**
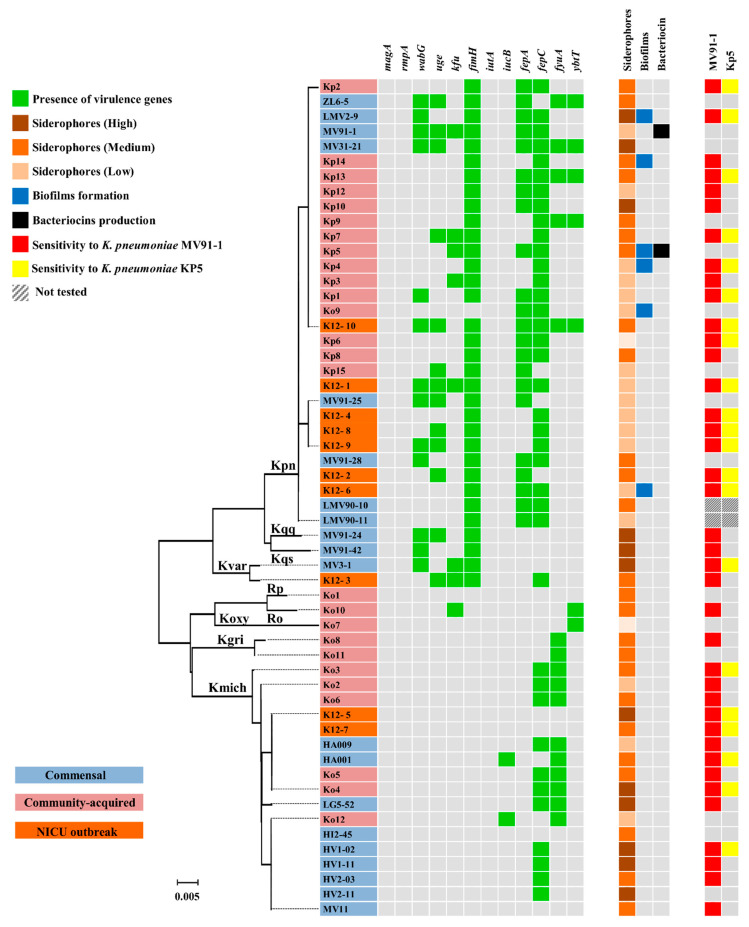
Virulence genes and phenotypic features of the *Klebsiella* isolates analysed in this study: The presence of genes involved in virulence, i.e., *magA* and *rmpA* (hypermucoviscosity phenotype), *wabG* (lipopolysaccharide synthesis), *uge* (uridine diphosphate galacturonate 4-epimerase), *kfu* (iron-uptake system), *fimH* (type 1 fimbrial adhesin), *iutA* (siderophore Aerobactin), *iut B* (receptor of Aerobactin), *fepA* (siderophore Enterobactin), *fepC* (receptor of Enterobactin), *fyuA* (siderophore Yersibactin), *ybtT* (receptor of Yersibactin), was determined by PCR. Production of siderophores was quantified in cell-free supernatants; biofilm formation was assayed and quantified in PVC microtiter plates; production of proteinaceous antimicrobial compounds (bacteriocins) was tested on solid culture medium against all *Klebsiella* isolates of this study, used as indicator strains. Sensitivity to bacteriocins produced by *K. pneumoniae* MV91-1 and Kp5 was assayed on solid medium.

**Figure 4 microorganisms-09-02344-f004:**
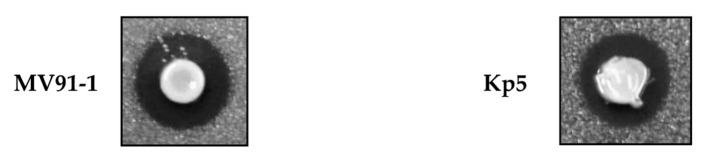
Antimicrobial activity of *K. pneumoniae* MV91-1 (commensal) and Kp5 (community-acquired) against *K. pneumoniae* K12-9, a multidrug ESBL-producer isolate from NICU.

**Table 1 microorganisms-09-02344-t001:** Biochemical and molecular identification (based on partial *rpoB* gene sequencing) of commensal, community-acquired, NICU outbreak and reference strains used in this study.

Strains	Biochemical ID	*rpoB* (% Identity)	Origin	Source *
**Commensal**				
HA001	*K. oxytoca*	*K. michiganensis* (100)	Faeces	UCM
HA009	*K. oxytoca*	*K. michiganensis* (100)	Faeces	UCM
HI2-45	*K. oxytoca*	*K. michiganensis* (100)	Faeces	UCM
HV1-02	*K. oxytoca*	*K. michiganensis* (99.52)	Faeces	UCM
HV1-11	*K. oxytoca*	*K. michiganensis* (99.52)	Faeces	UCM
HV2-03	*K. oxytoca*	*K. michiganensis* (99.52)	Faeces	UCM
HV2-11	*K. oxytoca*	*K. michiganensis* (99.52)	Faeces	UCM
LG5-52	*K. oxytoca*	*K. michiganensis* (99.52)	Milk	UCM
MV11	*K. oxytoca*	*K. michiganensis* (99.52)	Meconium	UCM
LMV2-9	*K. pneumoniae*	*K. pneumoniae* (99.76)	Milk	UCM
LMV6-5	*K. pneumoniae*	*K. pneumoniae* (99.76)	Milk	UCM
LMV90-10	*K. pneumoniae*	*K. pneumoniae* (100)	Milk	UCM
LMV90-11	*K. pneumoniae*	*K. pneumoniae* (100)	Milk	UCM
MV3-1	*K. pneumoniae*	*K. variicola* (100)	Faeces	UCM
MV31-21	*K. pneumoniae*	*K. pneumoniae* (99.76)	Faeces	UCM
MV91-1	*K. pneumoniae*	*K. pneumoniae* (99.76)	Faeces	UCM
MV91-24	*K. pneumoniae*	*K. quasipneumoniae* subsp. *similipneumoniae* (100)	Faeces	UCM
MV91-25	*K. pneumoniae*	*K. pneumoniae* (99.76)	Faeces	UCM
MV91-28	*K. pneumoniae*	*K. pneumoniae* (100)	Faeces	UCM
MV91-42	*K. pneumoniae*	*K. quasipneumoniae* subsp. *quasipneumoniae* (99.52)	Faeces	UCM
**Community-acquired**				
Ko1	*K. pneumoniae*	*R. planticola* (100)	Blood culture	RYC
Ko2	*K. oxytoca*	*K. michiganensis* (99.76)	Blood culture	RYC
Ko3	*K. oxytoca*	*K. michiganensis* (99.52)	Blood culture	RYC
Ko4	*K. oxytoca*	*K. michiganensis* (100)	Blood culture	RYC
Ko5	*K. oxytoca*	*K. michiganensis* (100)	Blood culture	RYC
Ko6	*K. oxytoca*	*K. michiganensis* (99.76)	Blood culture	RYC
Ko7	*K. oxytoca*	*K. oxytoca* (100)	Blood culture	RYC
Ko8	*K. oxytoca*	*K. grimontii* (98.80)	Blood culture	RYC
Ko9	*K. oxytoca*	*K. pneumoniae* (99.76)	Blood culture	RYC
Ko10	*K. oxytoca*	*R. ornithinolytica* (100)	Blood culture	RYC
Ko11	*K. oxytoca*	*K. grimontii* (99.04)	Blood culture	RYC
Ko12	*K. oxytoca*	*K. michiganensis* (99.52)	Blood culture	RYC
Kp1	*K. pneumoniae*	*K. pneumoniae* (99.76)	Blood culture	RYC
Kp2	*K. pneumoniae*	*K. pneumoniae* (99.52)	Blood culture	RYC
Kp3	*K. pneumoniae*	*K. pneumoniae* (99.76)	Blood culture	RYC
Kp4	*K. pneumoniae*	*K. pneumoniae* (99.76)	Blood culture	RYC
Kp5	*K. pneumoniae*	*K. pneumoniae* (99.76)	Blood culture	RYC
Kp6	*K. pneumoniae*	*K. pneumoniae* (100)	Blood culture	RYC
Kp7	*K. pneumoniae*	*K. pneumoniae* (99.76)	Blood culture	RYC
Kp8	*K. pneumoniae*	*K. pneumoniae* (100)	Blood culture	RYC
Kp9	*K. pneumoniae*	*K. pneumoniae* (99.76)	Blood culture	RYC
Kp10	*K. pneumoniae*	*K. pneumoniae* (99.76)	Blood culture	RYC
Kp12	*K. pneumoniae*	*K. pneumoniae* (99.76)	Blood culture	RYC
Kp13	*K. pneumoniae*	*K. pneumoniae* (99.76)	Blood culture	RYC
Kp14	*K. pneumoniae*	*K. pneumoniae* (99.76)	Blood culture	RYC
Kp15	*K. pneumoniae*	*K. pneumoniae* (100)	Blood culture	RYC
**NICU outbreak**				
K12-1	*K. pneumoniae*	*K. pneumoniae* (100)	Blood culture	HUDO
K12-2	*K. pneumoniae*	*K. pneumoniae* (100)	Blood culture	HUDO
K12-3	*K. pneumoniae*	*K. variicola* (99.52)	NICU environment	HUDO
K12-4	*K. pneumoniae*	*K. pneumoniae* (99.76)	NICU environment	HUDO
K12-5	*K. pneumoniae*	*K. michiganensis* (100)	Faeces	HUDO
K12-6	*K. pneumoniae*	*K. pneumoniae* (100)	Vascular catheter	HUDO
K12-7	*K. oxytoca*	*K. michiganensis* (100)	Faeces	HUDO
K12-8	*K. pneumoniae*	*K. pneumoniae* (99.76)	Faeces	HUDO
K12-9	*K. pneumoniae*	*K. pneumoniae* (99.76)	Faeces	HUDO
K12-10	*K. pneumoniae*	*K. pneumoniae* (99.76)	Faeces	HUDO
**Reference type strains (^T^)**				
DSM 30104^T^	*K. pneumoniae* subsp. *pneumoniae*	*K. pneumoniae* (100)	Unknown	DSMZ
CECT 142	*K. pneumoniae* subsp. *pneumoniae*	*K. pneumoniae* (99.76)	Unknown	CECT
CECT 517	*K. pneumoniae* subsp. *pneumoniae*	*K. pneumoniae* (100)	Urine	CECT
DSM 16231^T^	*K. pneumoniae* subsp. *rhinoscleromatis*	*K. pneumoniae* (100)	Nose rhinoscleroma	DSMZ
DSM 16358^T^	*K. pneumoniae* subsp. *ozaenae*	*K. pneumoniae* (100)	Nose	DSMZ
CECT 860^T^	*K. oxytoca*	*K. oxytoca* (100)	Pharyngeal tonsil	CECT

Abbreviations: * UCM, group 920080, Complutense University of Madrid; RYC, Hospital Universitario Ramón y Cajal; HUDO, Hospital Universitario 12 de Octubre; CECT, Spanish Type Culture Collection; DSMZ, German Collection of Microorganisms and Cell Cultures GmbH.

## Data Availability

The *rpoB* gene sequences obtained in this study were deposited in the GenBank database, under the accession numbers KJ499842 to KJ499903.

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
