# Peer review of "Phenotypic and Molecular Characterization of Commensal, Community-Acquired and Nosocomial Klebsiella spp."

_microorganisms, 2021, doi:10.3390/microorganisms9112344_

Round 1

Reviewer 1 Report

The paper is well written. There are several points that need to be addressed as follows:

Major comments:

  1. Abstract lacks the key message of the manuscript.
  2. What do the authors interpret from the presence of virulence traits in commensal strains? Why were they present and yet not causing disease? It should be discussed in more detail.

Minor comments:

  1. Line 12: change ‘treat’ to ‘threat’
  2. Line 54: far less studied ‘than’
  3. Were the infants, from whom faeces and meconium were collected, fed breast-milk that contained Klebsiella? Or did human milk (n=5) included milk from those infants’ mothers?
  4. Line 146: as recommended ‘by’ the manufacturer…
  5. Line 188: The ability ‘to’ inhibit the growth…

Author Response

Responses to Reviewer 1

Major comments:

  1. Abstract lacks the key message of the manuscript.

R1. We have added the following statement in the abstract:

“Globally, we report that commensal strains present virulence traits (virulence genes, siderophores and biofilms) comparable to community-acquired and NICU infective isolates, thus suggesting that the human microbiota could constitute a reservoir for infection”.

  1. What do the authors interpret from the presence of virulence traits in commensal strains? Why were they present and yet not causing disease? It should be discussed in more detail.

R2. That is a very interesting question but, unfortunately, the link between Klebsiella colonisation and progression to infection are currently poorly understood. However, we have discussed the possible origin of virulence and antibiotic-resistance genes in the discussion section, as follows:

“The presence of virulence traits within each commensal strain could be determined by some of the conserved genes that form the core genome within each Klebsiella species, while other virulence and antibiotic resistance genes would be determined by a pool of accessory genes (accessory genome) that can be shared between different Klebsiella species and even between different genera [58]. Nowadays, it is becoming more evident that Klebsiella has the ability of exchanging and assembling a wide portfolio of genes that are involved in colonization, infection and antimicrobial resistance, which in last instance would determine if a commensal strain remains asymptomatic or turns pathogenic [58]. Commensal Klebsiella isolates, which are well adapted to live and compete within the human microbiota, are prone to acquire antibiotic resistant genes, by horizontal gene transfer, after the colonization with antibiotic-resistant Klebsiella strains acquired in hospital environments, such as a NICU, or from other host bacteria that have developed resistance after antibiotic treatments [59].”

We have added the following references to the reference section:

  1. Martin, R.M.; Bachman, M.A. Colonization, Infection, and the Accessory Genome of Klebsiella pneumoniae. Front. Cell. Infect. Microbiol. 2018, 8, 4.
  2. Ghenea, A.E.; Cioboată, R.; Drocaş, A.I.; Țieranu, E.N.; Vasile, C.M.; Moroşanu, A.; Țieranu, C.G.; Salan, A.-I.; Popescu, M.; Turculeanu, A.; Padureanu, V.; Udriștoiu, A.-L.; Calina, D.; Cȃrţu, D.; Zlatian, O.M. Prevalence and Antimicrobial Resistance of Klebsiella Strains Isolated from a County Hospital in Romania. Antibiotics 2021, 10, 868.

Minor comments:

  1. Line 12: change ‘treat’ to ‘threat’

R1. Corrected

  1. Line 54: far less studied ‘than’: corrected

R2. Corrected.

  1. Were the infants, from whom faeces and meconium were collected, fed breast-milk that contained Klebsiella? Or did human milk (n=5) included milk from those infants’ mothers?

R3. We couldn’t determine if the milk from the infant’s mothers contained Klebsiella spp. However, we could determine the presence of DNA from Klebsiella pneumoniae and K. oxytoca in samples of breast milk, by DGGE, thus confirming the result of the cultures. Contrary to other genus, such as Staphylococcus, Lactobacillus and Bifidobacterium, where we have confirmed that breast milk and infant faeces from mother-infant pairs share the same strains, thus indicating that breastfeeding could contribute to the bacterial transfer from the mother to the infant and, therefore, to the infant gut colonization, Klebsiella was less represented in breast milk, being difficult to isolate.

To avoid confusion, we have modified the final paragraph of the introduction section as follows:

“In these studies, we obtained a collection of Klebsiella isolates from milk of healthy women and from meconium and faeces of breast-fed term infants”.

  1. Line 146: as recommended ‘by’ the manufacturer…

R4. Corrected.

  1. Line 188: The ability ‘to’ inhibit the growth…

R5. Corrected.

Reviewer 2 Report

General comments:

The article has an interesting idea because indeed most articles about Klebsiella’s resistance are using patient pathogenic strains. The authors used advanced methodology (DNA sequencing) for precise species identification and for detection of virulence traits, including antibiotic resistance genes.

  1. Authors should state an explanation in the Introduction section why breast-feeding woman are a representative sample of the general population, e.g. the majority of women will give birth at some point, and they need to got to hospital for that, etc.
  2. Authors should explain in the Discussion section how is possible that commensal Klebsiella strains to harbor resistance genes and virulence traits, e.g. Klebsiella is known as a resistance plasmids collectors, which are acquired from other host bacteria which developed resistance during antibiotic treatments (authors can add for example the following reference: Ghenea AE, Cioboată R, Drocaş AI, Țieranu EN, Vasile CM, Moroşanu A, Țieranu CG, Salan A-I, Popescu M, Turculeanu A, Padureanu V, Udriștoiu A-L, Calina D, Cȃrţu D, Zlatian OM. Prevalence and Antimicrobial Resistance of Klebsiella Strains Isolated from a County Hospital in Romania. Antibiotics. 2021; 10(7):868. https://doi.org/10.3390/antibiotics10070868). Another source are bacteriophages found in the water and food which harbor resistance genes, and these phages can come from example from chicken farms waste, because usually chickens are fed with antibiotics to grow faster, etc.
  3. Line 108: please add the period of incubation, e.g. 18/24h.
  4. Line 119: please add the name and producer of the sequencing machine used.
  5. Line 164: please replace “was done” by “was identified”.
  6. Line 256: rephrase as “subsequently confirmed by the double-disk synergy test, and presence of the 256 blaCTX-M-15 gene.”

Author Response

Responses to Reviewer 2

General comments:

The article has an interesting idea because indeed most articles about Klebsiella’s resistance are using patient pathogenic strains. The authors used advanced methodology (DNA sequencing) for precise species identification and for detection of virulence traits, including antibiotic resistance genes.

  1. Authors should state an explanation in the Introduction section why breast-feeding woman are a representative sample of the general population, e.g. the majority of women will give birth at some point, and they need to got to hospital for that, etc.

R1. We have added the following paragraph in the introduction section:

“Breastfeeding women are a representative sample of the general population since the majority of them will give birth at some point in a hospital, where the newborn will be ideally breastfeed within the first hour of birth”

  1. Authors should explain in the Discussion section how is possible that commensal Klebsiellastrains to harbor resistance genes and virulence traits, e.g. Klebsiellais known as a resistance plasmids collectors, which are acquired from other host bacteria which developed resistance during antibiotic treatments (authors can add for example the following reference: Ghenea AE, Cioboată R, DrocaÅŸ AI, Èšieranu EN, Vasile CM, MoroÅŸanu A, Èšieranu CG, Salan A-I, Popescu M, Turculeanu A, Padureanu V, UdriÈ™toiu A-L, Calina D, CȃrÅ£u D, Zlatian OM. Prevalence and Antimicrobial Resistance of Klebsiella Strains Isolated from a County Hospital in Romania. Antibiotics. 2021; 10(7):868. https://doi.org/10.3390/antibiotics10070868). Another source are bacteriophages found in the water and food which harbor resistance genes, and these phages can come from example from chicken farms waste, because usually chickens are fed with antibiotics to grow faster, etc.

R2. Thank yo very much for your suggestion. To accommodate the suggestions made by Reviewer 1 and Reviewer 2, we have included the following paragraph in the discussion section:

“The presence of virulence traits within each commensal strain could be determined by some of the conserved genes that form the core genome within each Klebsiella species, while other virulence and antibiotic resistance genes would be determined by a pool of accessory genes (accessory genome) that can be shared between different Klebsiella species and even between different genera [58]. Nowadays, it is becoming more evident that Klebsiella has the ability of exchanging and assembling a wide portfolio of genes that are involved in colonization, infection and antimicrobial resistance, which in last instance would determine if a commensal strain remains asymptomatic or turns pathogenic [58]. Commensal Klebsiella isolates, which are well adapted to live and compete within the human microbiota, are prone to acquire antibiotic resistant genes, by horizontal gene transfer, after the colonization with antibiotic-resistant Klebsiella strains acquired in hospital environments, such as a NICU, or from other host bacteria that have developed resistance after antibiotic treatments [59].”

We have added the following references to the reference section:

  1. Martin, R.M.; Bachman, M.A. Colonization, Infection, and the Accessory Genome of Klebsiella pneumoniae. Front. Cell. Infect. Microbiol. 2018, 8, 4.

  1. Ghenea, A.E.; Cioboată, R.; Drocaş, A.I.; Țieranu, E.N.; Vasile, C.M.; Moroşanu, A.; Țieranu, C.G.; Salan, A.-I.; Popescu, M.; Turculeanu, A.; Padureanu, V.; Udriștoiu, A.-L.; Calina, D.; Cȃrţu, D.; Zlatian, O.M. Prevalence and Antimicrobial Resistance of Klebsiella Strains Isolated from a County Hospital in Romania. Antibiotics 2021, 10, 868.

  1. Line 108: please add the period of incubation, e.g. 18/24h.

R3. Corrected.

  1. Line 119: please add the name and producer of the sequencing machine used.

R4. We have added the following information: “ABI Prism 3730; Applied Biosystems, Foster City, CA, USA”.

  1. Line 164: please replace “was done” by “was identified”.

R5. Corrected

  1. Line 256: rephrase as “subsequently confirmed by the double-disk synergy test, and presence of the 256 blaCTX-M-15 gene.”

R6. Rephrased as follows:  “subsequently confirmed by the double-disk synergy test, and presence of the blaCTX-M-15 gene”